# Microstructural asymmetries of the planum temporale predict functional lateralization of auditory-language processing

Peipei Qin[1†], Qiuhui Bi[2†], Zeya Guo[1], Liyuan Yang[1], Haokun Li[1], Peng Li[1], Xinyu Liang[1], Junhao Luo[1], Xiangyu Kong[1], Yirong Xiong[1], Bo Sun[2], Sebastian Ocklenburg[3,4,5], Gaolang Gong[1,6,7]*

[1]State Key Laboratory of Cognitive Neuroscience and Learning & IDG/McGovern Institute for Brain Research, Beijing Normal University, Beijing, China; [2]School of Artificial Intelligence, Beijing Normal University, Beijing, China; [3]Department of Psychology, Medical School Hamburg, Hamburg, Germany; [4]ICAN Institute for Cognitive and Affective Neuroscience, Medical School Hamburg, Hamburg, Germany; [5]Institute of Cognitive Neuroscience, Biopsychology, Faculty of Psychology, Ruhr University Bochum, Bochum, Germany; [6]Beijing Key Laboratory of Brain Imaging and Connectomics, Beijing Normal University, Beijing, China; [7]Chinese Institute for Brain Research, Beijing, China

*For correspondence:
gaolang.gong@bnu.edu.cn

[†]These authors contributed equally to this work

## eLife Assessment

The authors studied the relationship between structural and functional lateralization in the planum temporale region of the brain, whilst also considering the morphological presentation of a single or duplicated Heschl's gyrus. The analyses are **compelling** due to a large sample size, inter-rater reliability, and corrections for multiple comparisons. The associations in this **important** work might serve as a reference for future targeted-studies on brain lateralization.

**Abstract** Structural hemispheric asymmetry has long been assumed to guide functional lateralization of the human brain, but empirical evidence for this compelling hypothesis remains scarce. Recently, it has been suggested that microstructural asymmetries may be more relevant to functional lateralization than macrostructural asymmetries. To investigate the link between microstructure and function, we analyzed multimodal MRI data in 907 right-handed participants. We quantified structural asymmetry and functional lateralization of the planum temporale (PT), a cortical area crucial for auditory-language processing. We found associations between PT functional lateralization and several structural asymmetries, such as surface area, intracortical myelin content, neurite density, and neurite orientation dispersion. The PT structure also showed hemispheric-specific coupling with its functional activity. All these functional-structural associations are highly specific to within-PT functional activity during auditory-language processing. These results suggest that structural asymmetry underlies functional lateralization of the same brain area and highlights a critical role of microstructural PT asymmetries in auditory-language processing.

## Introduction

One compelling notion regarding brain asymmetry is that 'functional lateralization is guided by structural asymmetry,' hypothesizing a critical role of structural asymmetries in the neuroanatomical basis of functional lateralization (*Ocklenburg and Güntürkün, 2018*; *Wada, 2009*). As a well-known cortical area posterior to the Heschl's gyrus (HG), which includes most of the primary auditory cortex, the PT putatively plays an essential role in the functional lateralization of language processing (*Albouy et al., 2020*; *Galaburda et al., 1978*; *Hickok and Poeppel, 2000*; *Hickok and Poeppel, 2007*; *Moffat et al., 1998*; *Price, 2010*; *Steinmetz, 1996*; *Tzourio-Mazoyer et al., 2018*), and its lateralization in functional activation during auditory- or language-related tasks has been well demonstrated (*Albouy et al., 2020*; *Hunter et al., 2003*; *Pahs et al., 2013*; *Shapleske et al., 1999*; *Zatorre et al., 2002*). Additionally, the PT shows distinct leftward asymmetries in a variety of structural aspects, e.g., length, gray matter concentration/volume, surface area, cortical thickness, and columnar neuronal units, although the results might conflict between studies (*Anderson et al., 1999*; *Bloom et al., 2013*; *Chance et al., 2008*; *Dorsaint-Pierre et al., 2006*; *Eckert et al., 2006*; *Fukutomi et al., 2018*; *Galuske et al., 2000*; *Greve et al., 2013*; *Josse et al., 2003*; *Knaus et al., 2006*; *Moffat et al., 1998*; *Ocklenburg et al., 2018*; *Schmitz et al., 2019*; *Sigalovsky et al., 2006*; *Tzourio-Mazoyer et al., 2019*). Due to the role of the PT in language processing, the idea that these asymmetries may be related to leftward functional lateralization in speech processing is highly intuitive. Unexpectedly, examining the functional association of PT macrostructural asymmetries at the individual level almost always showed negative results, except for only a couple of studies demonstrating correlations between PT asymmetry of surface area and some non-PT-related indices of language lateralization, e.g., the lateralization of dichotic listening (*Guadalupe et al., 2022*; *Hugdahl et al., 2003*; *Dos Santos Sequeira et al., 2006*).

Several factors possibly account for such an unexpected majority of negative results. First, the sample size in previous studies is too small for such association analyses across individuals, compared to the recently suggested thousands of individuals for capturing a reproducible brain-wide association (*Marek et al., 2022*). Next, structural asymmetries were confined to PT macrostructural measures that are biologically nonspecific, e.g., surface area and thickness. Recent studies have used novel MRI techniques to quantify PT microstructure on the level of axons or dendrites and further revealed PT asymmetries in myelin content, neurite density, and neurite orientation dispersion (*Fukutomi et al., 2018*; *Ocklenburg et al., 2018*; *Schmitz et al., 2019*; *Sigalovsky et al., 2006*). Importantly, postmortem work on the microstructure of the temporal cortex has led to the hypothesis that asymmetries in the organization of the intrinsic microcircuitry of the PT and other areas may be crucial for functional lateralization (*Galuske et al., 2000*). In line with this idea, asymmetries in PT microstructure have been linked to electrophysiological correlates of auditory speech, an indirect measure of functional language lateralization (*Ocklenburg et al., 2018*). However, the crucial experiment would be to show a direct link between PT microstructure and PT functional lateralization measured with fMRI. Finally, duplicated HG (dHG) in the left or right hemisphere occurs substantially in healthy individuals, although less frequently than single HG (sHG) (*Campain and Minckler, 1976*; *Leonard et al., 1998*; *Marie et al., 2015*). In particular, such an HG gyrification pattern is accompanied by significant differences in PT structural properties and their asymmetries compared with the single HG (*Tzourio-Mazoyer and Mazoyer, 2017*). The HG gyrification pattern could further relate to the functional-structural association of PT asymmetries, therefore, confounding the interindividual correlation between PT asymmetries. Thus, the interindividual variation in the HG gyrification pattern should be taken into account when investigating the functional-structural association of PT asymmetries, but this has been largely overlooked in previous studies.

In the present study, we included a large cohort of 907 right-handed healthy young adults to assess whether and how PT lateralization of speech-related functional activation relates to its underlying structural asymmetries. Particularly, we chose to manually delineate bilateral PTs on the virtually reconstructed cortical surface for each individual, which is very labor intensive but can localize such an anatomically highly variable structure with minimized errors. Moreover, a variety of macrostructural and microstructural measures were applied to quantify PT structural asymmetries, including surface area, cortical thickness, myelin content, neurite density index (NDI), and orientation dispersion index (ODI). Here, cortical myelin content mainly reflects the degree of intracortical axonal myelination and was measured using a noninvasive T1w/T2w mapping approach (*Glasser et al., 2016*; *Glasser and Van Essen, 2011*). Cortical NDI and ODI mainly represent the density of intracortical neurite and the

complexity of dendritic arborization, respectively (*Ocklenburg et al., 2018*), and were estimated using the diffusion MRI-based neurite orientation dispersion and density imaging (NODDI) model (*Zhang et al., 2012*). Using these data, we evaluated (1) the functional and structural PT asymmetries at the group level, (2) the functional-structural coupling of PT asymmetries at the individual level, and (3) the within-hemispheric functional-structural coupling of PT at the individual level and its asymmetry. In these analyses, the influence of the HG gyrification pattern was considered.

## Results

Two experienced raters manually delineated PT and determined the HG gyrification pattern for 907 healthy right-handed young adults (*Figure 1*), with each rater assessing about half of the subjects.

As shown in *Table 1*, the manual operation showed excellent interrater reliability and imaging test-retest reproducibility for all PT measures. Among these young adults, left and right HG duplications were identified in 229 and 263 subjects, respectively (occurrence rate: left, 25.3%, right, 29.0%). In terms of the HG gyrification pattern in both hemispheres, all subjects were divided into four groups: 503 individuals with bilateral sHGs (L1/R1, 55.5%), 175 individuals with left sHG but right dHG (L1/R2, 19.3%), 141 individuals with left dHG but right sHG (L2/R1, 15.6%), and 88 individuals with bilateral dHGs (L2/R2, 9.7%). Age and sex did not differ among these four groups (*Table 2*).

The differences in age and sex distribution across the four groups were evaluated using one-way ANOVA and the Kruskal-Wallis test, respectively. HG, Heschl's gyrus. L1/R1, single HG on the left and single HG on the right; L1/R2, single HG on the left and duplicated HG on the right; L2/R1, duplicated HG on the left and single HG on the right; L2/R2, duplicated HG on the left and duplicated HG on the right. F, female; M, male.

For each subject, fMRI activation of the left and right PT during an auditory-language comprehension task was estimated. Both left and right PTs showed significant group-level activation for all three contrasts (i.e. 'story – baseline,' 'math – baseline,' and 'story – math') within the task (*Figure 2—figure supplement 1*), indicating a strong functional involvement of both PTs in such auditory-language processing. Factor analysis of the activation T values of the three contrasts revealed two factors in each hemisphere: one representing PT functional activation of speech perception and the other representing PT functional activation of speech comprehension.

We evaluated the correlation of functional and structural measures of the left and right PT with two available behavioral language test scores (the oral reading recognition test, measuring the ability to read decoding; the picture vocabulary test, measuring the ability of linguistic comprehension), but found no significant results (all $P_{FWE} > 0.05$).

### PT functional and structural asymmetries at the group level

For each of the four groups above, PT functional lateralization of speech perception and comprehension activation were evaluated while controlling for age, gender, and brain size. As shown in *Figure 2*, significant leftward PT lateralization was observed for both speech perception and comprehension in the L1/R1 (Cohen's d: speech perception = 0.35, speech comprehension = 0.36) and L1/R2 (Cohen's d: speech perception = 0.52, speech comprehension = 0.57) groups but not in the L2/R1 and L2/R2 groups (*Supplementary file 1a*), suggesting a specific influence of the HG gyrification pattern on speech-related functional lateralization of the PT. For each hemisphere, we further compared these PT activations between subjects with sHG and dHG. The results showed that HG duplication was largely accompanied by decreased functional activation in the ipsilateral PT, and the degree of such a decrease varied between the left and right PTs (*Figure 2—figure supplement 2*).

Similar analyses were applied to PT structural measures, i.e., surface area, thickness, myelin content, NDI, and ODI. As shown in *Figure 3*, ODI and cortical thickness showed consistent leftward and rightward asymmetry across all 4 groups, suggesting a minimal influence of the HG gyrification pattern on the group-level PT asymmetry of these particular measures. In contrast, there was the more or less confounding influence of the HG gyrification pattern on the PT asymmetry of the other four structural measures. Specifically, we observed (1) significant leftward asymmetry of PT surface area in the L1/R1, L1/R2, and L2/R2 groups (Cohen's d: L1/R1 = 0.42, L1/R2 = 1.48, L2/R2 = 0.73) but no asymmetry in the L2/R1 group; (2) significant leftward and rightward asymmetry of PT myelin content only in the L1/R2 and L2/R1 groups (Cohen's d: L1/R2 = 0.58, L2/R1 = –0.68); and (3) significant leftward NDI

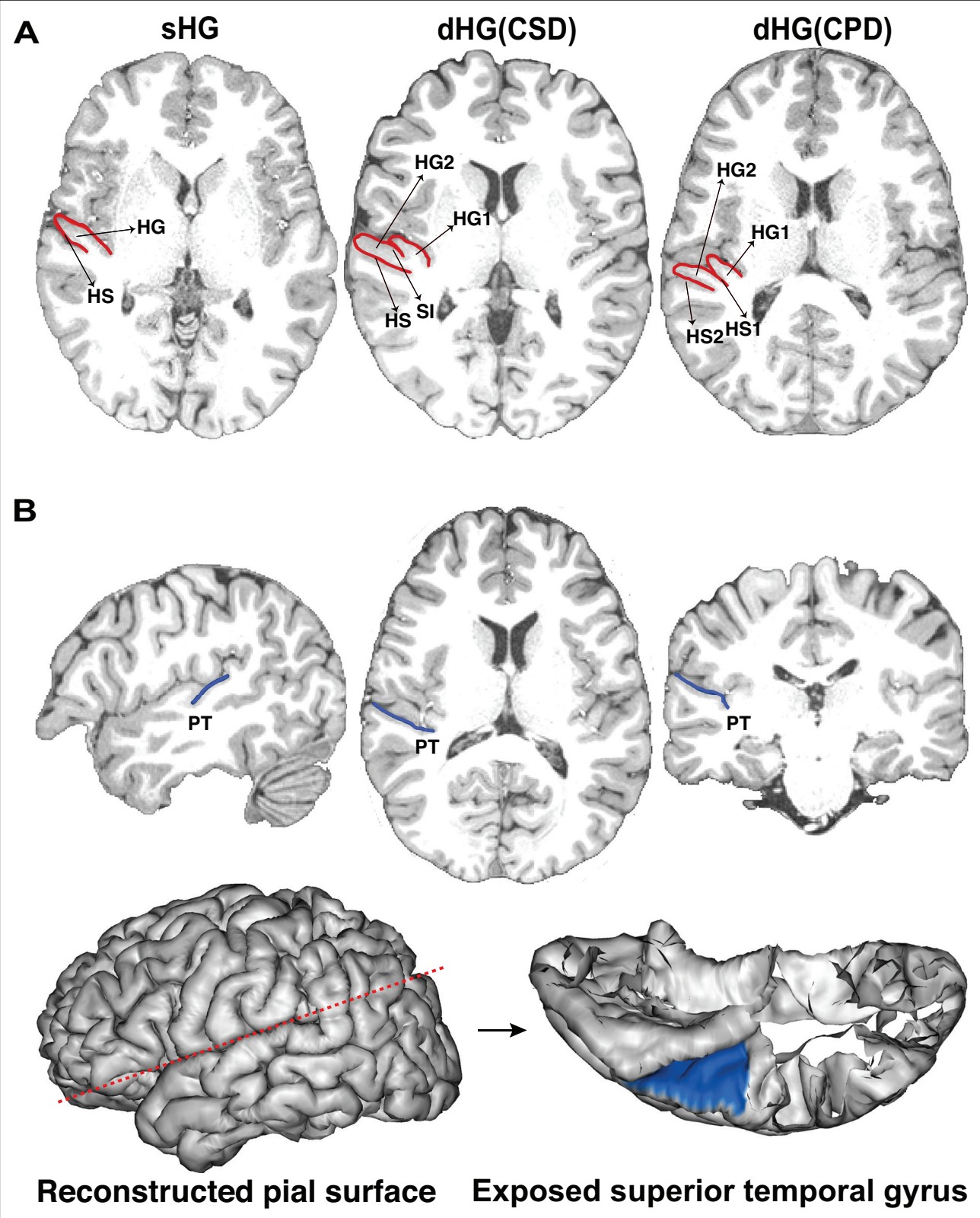

**Figure 1.** The gyrification pattern of Heschl's gyrus (HG) and manual delineation of the planum temporale (PT) on individual images. (**A**) Examples of three types of HG gyrification patterns (red line): are single HG (sHG), common stem duplication (CSD), and complete duplication (CPD). HG1, the anterior part of duplicated HG; HG2, the posterior part of duplicated HG; SI, the sulcus intermedius; HS1, the anterior Heschl's sulcus; HS2, the posterior Heschl's sulcus. (**B**) An example of manually delineated PT on a T1-weighted image (blue line) and reconstructed 32k~ pial surface (blue area). The

*Figure 1 continued on next page*

*Figure 1 continued*

oblique section plane (red dotted line) was applied to expose the internal surface of the superior temporal gyrus using the Anatomist (*Rivière et al., 2011*).

asymmetry in the L1/R2 group (Cohen's d: L1/R2 = 0.40) but rightward NDI asymmetry in both the L1/R1 and L2/R1 groups (Cohen's d: L1/R1 = –0.21, L2/R1 = –0.48) (*Supplementary file 1a*). In addition, the comparison of these PT structural measures between subjects with sHG and dHG indicated that HG duplication was accompanied by a decrease in surface area, myelin content, NDI, and ODI of the ipsilateral PT, with the left and right PT also showing variable degrees of such a decrease (*Figure 3—figure supplement 1*).

We also evaluated whether functional and structural PT asymmetries correlate with the two behavioral language test scores. The AI of PT speech comprehension activation was found significantly correlated with picture vocabulary test scores ($R = 0.10$, $P_{FWE} = 0.047$). No any other correlation was observed.

## Structure-function associations of PT asymmetries at the individual level

For each functional or structural PT measure, an asymmetry index (AI) was calculated for each subject. For each functional-structural pair of PT measures (2 × 5 pairs in total), we evaluated the interindividual correlation of their AIs with consideration of the HG gyrification pattern (including a group factor, i.e., L1/R1, L1/R2, L2/R1, or L2/R2), while controlling for age, gender, and brain size. No significant group effect (i.e. the interaction with the HG gyrification pattern in the general linear model) was observed for any functional-structural AI pair.

As shown in *Figure 4*, among these functional-structural pairs, the functional AI of speech perception activation showed significant positive correlations with the AIs of myelin content ($R = 0.26$, $P_{FWE} = 1.22 \times 10^{-13}$), NDI ($R = 0.13$, $P_{FWE} = 2.00 \times 10^{-3}$), and ODI ($R = 0.22$, $P_{FWE} = 1.75 \times 10^{-9}$), regardless of the HG gyrification pattern. In contrast, functional AI of speech comprehension activation significantly correlated with the AIs of surface area ($R = 0.21$, $P_{FWE} = 2.66 \times 10^{-9}$), myelin content ($R = 0.20$, $P_{FWE} = 4.66 \times 10^{-8}$), and NDI ($R = 0.11$, $P_{FWE} = 1.47 \times 10^{-2}$), regardless of the HG gyrification pattern. Notably, the observed interindividual correlations were consistently positive, strongly supporting an individual-level coupling of PT functional and structural asymmetries.

## Within-hemispheric PT structure-function associations at the individual level

In addition to PT functional and structural AIs, we evaluated the correlations between PT functional and structural measures of each hemisphere separately, while controlling for age, gender, and brain size (*Figure 5*). For speech perception, functional activation correlated positively with myelin content and ODI for either the left (myelin content: $R = 0.16$, $P_{FWE} = 6.47 \times 10^{-5}$; ODI: $R = 0.17$, $P_{FWE} = 2.88 \times 10^{-5}$) or right PT (myelin content: $R = 0.20$, $P_{FWE} = 1.30 \times 10^{-7}$; ODI: $R = 0.18$, $P_{FWE} = 7.18 \times 10^{-6}$),

**Table 1.** The reliability of planum temporale (PT) measures from manually delineated PT.

| Reliability | | Interrater ICC (N=20) | Imaging test-retest ICC (N=43) |
|---|---|---|---|
| | Speech perception | 0.99 | 0.55 |
| PT functional activation | Speech comprehension | 0.99 | 0.78 |
| | Surface area | 0.85 | 0.69 |
| | Thickness | 0.97 | 0.72 |
| | Myelin content | 0.98 | 0.61 |
| | NDI | 0.99 | 0.73 |
| PT structural measures | ODI | 0.96 | 0.86 |

ICC, intraclass correlation coefficient; PT, planum temporale; NDI, neurite density index; ODI, orientation dispersion index.

**Table 2.** The distribution of Heschl's gyrus (HG) gyrification patterns.

| | Type of HG gyrification pattern | | | | |
| --- | --- | --- | --- | --- | --- |
| | L1/R1 | L1/R2 | L2/R1 | L2/R2 | Group difference |
| Subject number | 503 | 175 | 141 | 88 | |
| Age (mean±SD) | 28.8±3.7 | 28.8±3.8 | 29.2±3.6 | 28.1±3.6 | p=0.19 |
| Sex (F/M) | 275/228 | 96/79 | 83/58 | 55/33 | p=0.49 |

regardless of the HG gyrification pattern. The permutation test, however, showed no significant difference in the degree of functional-structural correlations between the left and right PTs. Moreover, speech perception activation showed significant correlations with surface area ($R = –0.11$, $P_{FWE} = 2.33 \times 10^{-2}$) and NDI ($R = 0.14$, $P_{FWE} = 2.16 \times 10^{-3}$) for the right PT but not the left PT. Regarding speech comprehension, functional activation significantly correlated with myelin content for the right PT ($R = 0.16$, $P_{FWE} = 2.59 \times 10^{-5}$) but not the left PT. We also observed a significant correlation between functional activation of speech comprehension and NDI for the left PT ($R = –0.12$, $P_{FWE} = 1.62 \times 10^{-2}$) but not the right PT.

## Specificity of the observed PT structure-function associations

For the included subjects, another six task-based fMRI scans were acquired. To test whether the observed PT functional-structural couplings above are specific to auditory-language processing, we further estimated left and right PT activation of the main contrast from each of these tasks as well as their AI of PT activation, and then repeated all PT coupling analyses for these tasks. As shown in *Supplementary file 1b-1e*, there were only very few significant correlations among the PT functional-structural pairs, therefore, supporting the specificity of our currently observed PT functional-structural couplings to the functional activation of auditory-language processing.

In addition, to evaluate the spatial specificity of our observed PT functional-structural couplings, we estimated the functional activation of speech perception and comprehension of the entire hemisphere from the main auditory-language processing task. The coupling analysis showed no significant interindividual correlation between PT structural AIs and speech-related functional AIs of the entire hemisphere or between PT structural measures and speech-related functional activation of the entire

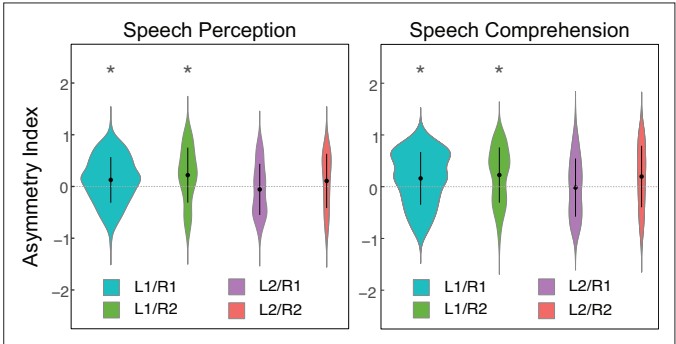

**Figure 2.** Planum temporale (PT) functional lateralization of speech processing at the group level. For each group, a linear mixed model (LME) was used to test the hemispheric asymmetry of each PT functional activation. 'Hemisphere' was the fixed effect, asymmetry index was the response variable, age, and sex, and total brain volume were covariates. L1/R1, single HG on the left and single HG on the right; L1/R2, single HG on the left and duplicated HG on the right; L2/R1, duplicated HG on the left and single HG on the right; L2/R2, duplicated HG on the left and duplicated HG on the right. * denotes a significant difference between left and right PTs (i.e. $P_{FWE}$ <0.05). Positive and negative values of the asymmetry index represent leftward and rightward asymmetry, respectively.

The online version of this article includes the following figure supplement(s) for figure 2:

**Figure supplement 1.** fMRI activation maps of the human connectome project (HCP) language processing task.

**Figure supplement 2.** The difference in planum temporale (PT) functional activation between subjects with single and duplicated Heschl's gyrus (HG).

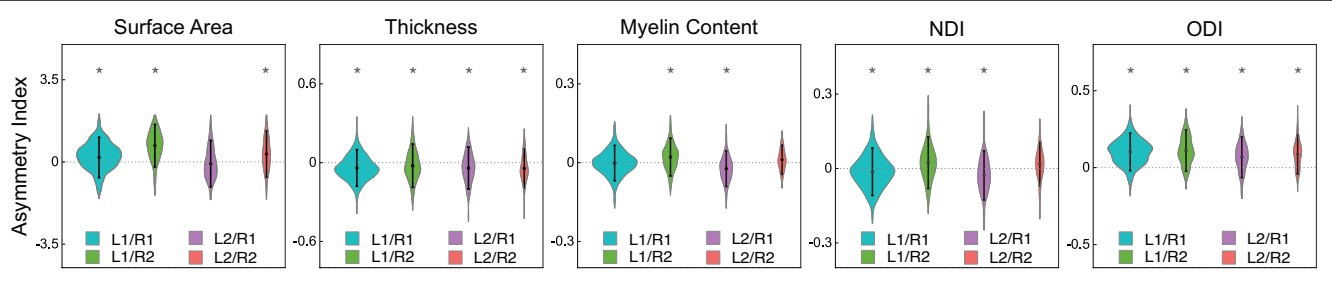

**Figure 3.** planum temporale (PT) structural asymmetry at the group level. For each group, a linear mixed model (LME) was used to test the hemispheric asymmetry of each PT structural measure. "Hemisphere" was the fixed effect, and the asymmetry index was the response variable. Age, sex, and total brain volume were covariates. Notably, we did not standardize these structural measures, so the scales differed between indicators. L1/R1, single HG on the left and single HG on the right; L1/R2, single HG on the left and duplicated HG on the right; L2/R1, duplicated HG on the left and single HG on the right; L2/R2, duplicated HG on the left and duplicated HG on the right. NDI, neurite density index; ODI, orientation dispersion index. * denotes a significant difference between left and right PTs ($P_{FWE}$ <0.05). Positive and negative values of the asymmetry index represent leftward and rightward asymmetry, respectively.

The online version of this article includes the following figure supplement(s) for figure 3:

**Figure supplement 1.** The difference in planum temporale (PT) structural measures between subjects with single and duplicated Heschl's gyrus (HG).

ipsilateral hemisphere (*Supplementary file 1f-1i*). Therefore, PT structural measures or their asymmetries were coupled with PT-specific functional activation or their lateralization rather than with general functional activation or its lateralization in auditory-language processing.

## Discussion

Using a large cohort of healthy adults with high-quality multimodal MRI data and highly accurate PT determination, the present study shows clear associations between structural and functional asymmetries in the PT. The observed functional-structural associations of PT asymmetries are highly specific to the within-PT functional activation of auditory-language processing.

As one of the most prominent structural asymmetries across the entire human brain, PT structural asymmetries have been well recognized and widely believed to play a critical role in human auditory

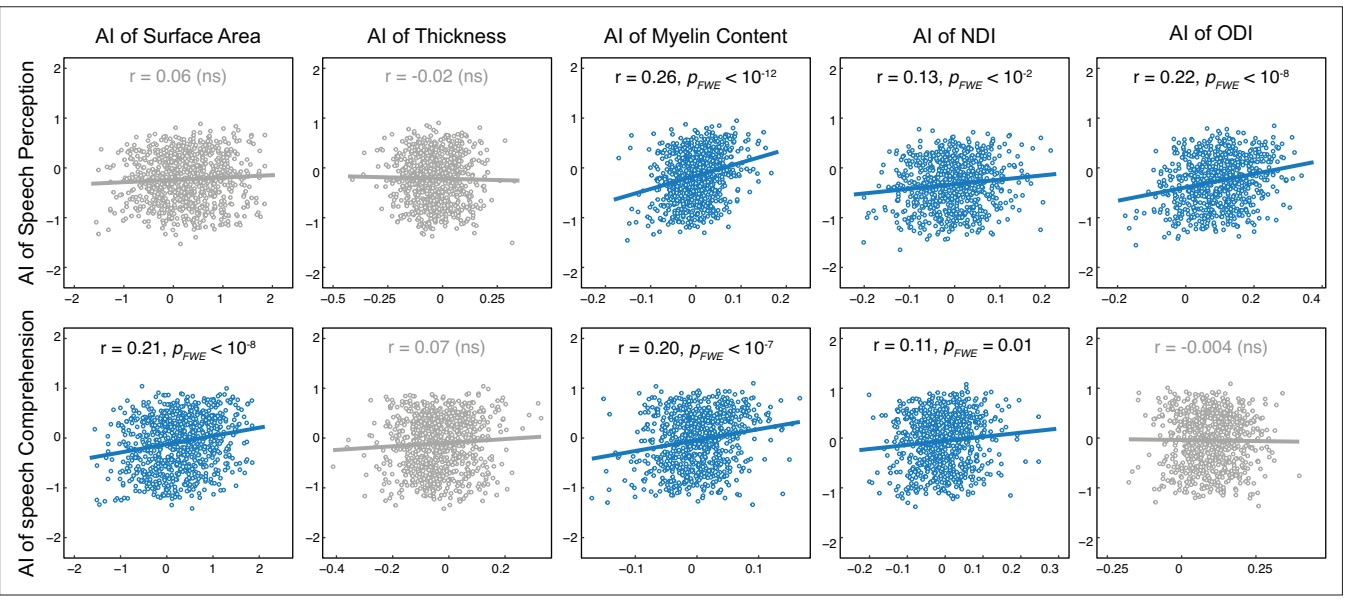

**Figure 4.** Interindividual correlation of planum temporale (PT) functional and structural asymmetries. For each pair of correlations, a general linear model (GLM) was used. 'Functional AI' was the response variable and 'structural AI,' 'group,' and 'structural AI × group' were predictor variables. Age, sex, and total brain volume were covariates. The scatter plots for nonsignificant correlations ($P_{FWE}$ >0.05) are colored gray. AI, asymmetry index; NDI, neurite density index; ODI, orientation dispersion index; ns, non-significant.

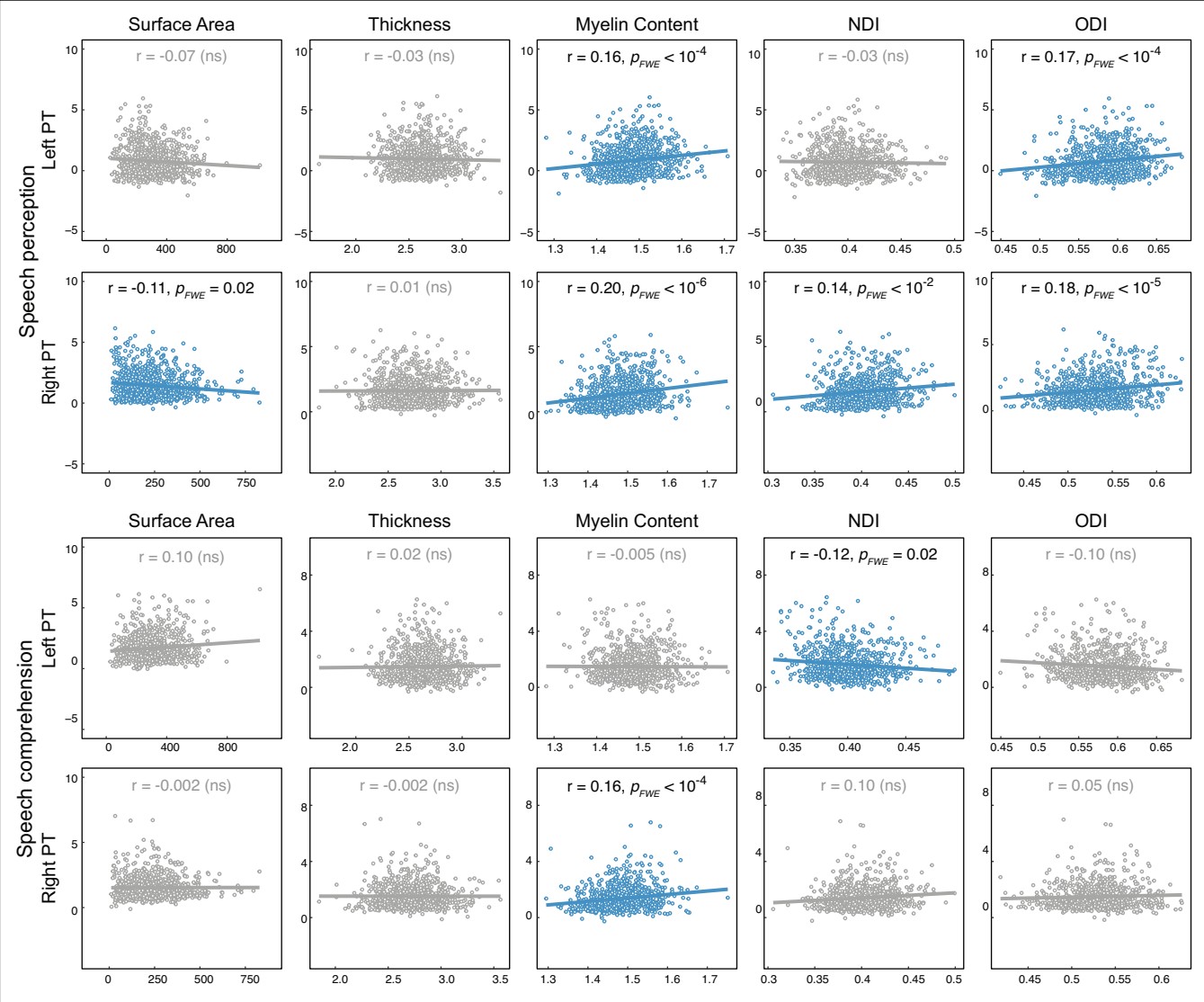

**Figure 5.** Interindividual functional-structural correlations of the left and right planum temporale (PT). For each pair of correlations, a general linear model (GLM) was used. 'Functional measure' was the response variable and 'structural measure,' 'group,' and 'structural measure × group' were predictor variables. Age, sex, and hemispheric brain volume were covariates. The scatter plots for nonsignificant correlations ($P_{FWE}$ >0.05) are colored gray. NDI, neurite density index; ODI, orientation dispersion index; ns, non-significant.

or language processing (*Bloom et al., 2013*; *Eckert and Leonard, 2000*; *Keller et al., 2019*; *Ocklenburg et al., 2018*; *Steinmetz, 1996*; *Yuan et al., 2021*; *Zatorre et al., 2002*). The anatomical leftward asymmetry of PT emerges at an early stage of life, prior to any environmental influences such as language exposure (*Dubois et al., 2010*; *Witelson and Pallie, 1973*). This provides evidence of an early functional lateralization in the speech-processing capacities of the two hemispheres (*Dehaene-Lambertz et al., 2002*). In concordance, anomalies of PT asymmetries in surface area or thickness have been reported in various brain disorders with auditory or language deficits, e.g., dyslexia, autism, and schizophrenia (*Altarelli et al., 2014*; *Bloom et al., 2013*; *Floris et al., 2016*; *Hynd and Semrud-Clikeman, 1989*; *Knaus et al., 2018*; *Pahs et al., 2013*; *Sumich et al., 2002*; *Sumich et al., 2005*; *Vanderauwera et al., 2018*), suggesting that asymmetric PT anatomy was an optimal architecture for the setting up of efficient speech networks. Moreover, there exists evolutionary evidence supporting the role of the PT as an anatomical substrate for language lateralization. For example, the leftward structural asymmetry of the PT has been observed in multiple non-human primates, including chimpanzees, macaques, and baboons (*Becker et al., 2024*; *Gannon et al., 1998*; *Xia et al., 2020*). Particularly, recent studies on baboons further demonstrated that PT structural leftward asymmetry in

newborn baboons could predict future development of communicative gestures, implying a key role of PT structural asymmetry in the lateralized communication system for human and non-human brain evolution (*Becker et al., 2021*; *Becker et al., 2024*).

To understand the role of PT structural asymmetries in language, it is critical to determine how they relate to functional patterns of language-related processing, e.g., language-related functional lateralization. Some evidence has demonstrated an association of PT structural asymmetry with language-related functional lateralization, but these functional lateralization were not measured directly from the PT (*Greve et al., 2013*; *Josse et al., 2003*; *Seghier et al., 2011*; *Tzourio-Mazoyer et al., 2018*). For instance, PT asymmetry in surface area was correlated with regional lateralization of language-related functional activation around the Sylvian fissure but not the activation of the PT (*Tzourio-Mazoyer et al., 2018*). Our present study provides the first empirical evidence of interindividual associations between functional and structural asymmetries within the PT, implying a functional pathway from PT structural asymmetry to PT functional lateralization to effective speech processing. A large sample size and accurate PT localization based on the well-trained manual operation were crucial in obtaining these results. Our present manually delineating method should be able to measure the PT more reliably and accurately, compared with previous atlas-based methods (*Chai et al., 2021*; *Kuiper et al., 2020*; *Ocklenburg et al., 2018*). The previously observed negative results of structural-functional correlation for PT asymmetries might partly be attributed to coarse locating of PT regions (*Greve et al., 2013*; *Liem et al., 2014*; *Tzourio-Mazoyer et al., 2018*). For PT-related studies with large sample sizes, however, it is difficult to apply such a labor-intensive manual PT labeling approach, and therefore, an accurate automatic labeling approach needs to be developed. Notably, our observed PT functional-structural associations were across healthy right-handed young adults, and it is unclear whether these associations could be extrapolated to children, left-handed adults, or patients. Therefore, future studies are required to evaluate the functional-structural association of PT asymmetries in other populations, which should provide insight into the modulating factor of functional-structural associations of PT.

In the context of brain asymmetries, our currently observed functional-structural coupling of PT asymmetries at the individual level empirically proves the role of structural gray matter asymmetries in the functional lateralization of the same brain area, supporting the compelling hypothesis that 'functional lateralization is guided by structural asymmetry' (*Wada, 2009*). For a specific brain area, however, its gray matter asymmetries are unlikely to be the only determinant for its functional lateralization. The relevant corpus callosum (interhemispheric connection) and structural white matter asymmetries should also play important roles in its functional lateralization, and integrating the three factors is encouraged for detangling structural mechanisms underlying functional lateralization (*Allendorfer et al., 2016*; *Catani et al., 2005*; *Catani et al., 2007*; *Ocklenburg et al., 2013*; *Ocklenburg and Güntürkün, 2018*).

The observed functional-structural coupling of PT asymmetries strongly depends on functional and structural measures. On the functional side, only PT functional activation of auditory speech processing showed such coupling with PT structural asymmetries, indicating the functional and spatial specificity of PT lateralization in auditory-language processing (*Forseth et al., 2020*; *Gernsbacher and Kaschak, 2003*; *Hickok and Poeppel, 2000*; *Steinmetz, 1996*). Speech perception and comprehension also showed functional-structural coupling with different structural measures, compatible with the dissociation of these two speech-processing components (*Dehaene-Lambertz et al., 2005*; *Price, 2010*). It is possible that hemispheric lateralization of different functional processing are generally related to distinct structural mechanisms. However, it should be noted that the fMRI language task used in the present study was not specifically designed to study PT activation and its lateralization (*Binder et al., 2011*), and therefore, the observed functional lateralization of speech perception and speech comprehension in the PT was not measured directly by a specific language task. Functional lateralization estimated indirectly by a data-driven approach is less specific in nature, which may partly account for the modest effect size of our observed functional-structural correlations. This also makes it difficult to further interpret the functional-structural correlation's differential pattern observed between speech perception and speech comprehension. Similarly, our specificity analyses were not based on other specifically designed language tasks for PT-associated non-speech auditory processing (e.g. reading or word production) but on available task-fMRI data from the HCP. Future investigations with more specifically designed fMRI tasks on PT activation are warranted to verify and extend the

PT functional-structural coupling results. On the structural side, the observed functional-structural coupling of PT asymmetries mainly involves microstructural measures, e.g., myelin content, neurite density, and neurite orientation dispersion, except for the association of speech comprehension with surface area. These microstructural measures represent specific aspects of cortical composition on the level of axons or dendrites (*Glasser and Van Essen, 2011*; *Zhang et al., 2012*), therefore, implying a dominant role of within-PT microconnectional or microcircuitry asymmetries in functional asymmetries of PT. Specific microstructural asymmetry between bilateral PTs might cause less recruitment of the nondominant hemisphere in particular speech processing, therefore, resulting in more pronounced functional lateralization (*Galaburda et al., 1990*; *Tzourio-Mazoyer et al., 2018*). Another possible explanation could be that higher myelin content and larger surface area in left PT potentially indicated more white matter connection with other language-related regions such as Broca's area, and therefore, is more involved in language tasks than its right homolog (*Allendorfer et al., 2016*; *Catani et al., 2005*; *Giampiccolo and Duffau, 2022*).

Compatibly with the functional-structural coupling of PT asymmetries, most microstructural measures, including myelin content, neurite density, and neurite orientation dispersion, showed an association with speech-related activation for the left or right PT per se. Similar functional-structural associations have been observed in other non-PT cortical areas, e.g., between myelin content and task-evoked activities (*Glasser et al., 2014*; *Grydeland et al., 2016*; *Helbling et al., 2015*; *Hunt et al., 2016*; *Kim and Knösche, 2016*; *Ma and Zhang, 2017*). These findings together indicate an important contribution of intracortical microcircuits to functional activity: cortical areas with greater intracortical fiber density and more complex dendritic structures are likely accompanied by stronger local activation during the task (*Hall et al., 2003*; *Hunt et al., 2016*). As for macrostructural measures, the asymmetric PT surface area was also associated with speech comprehension AI. Given that the within-hemispheric coupling tendency between surface and speech comprehension existed only in the left PT, it was possible that the larger surface area of the left PT led to a less recruitment of its right homologous, and therefore, the lateralization of functional activity would be more pronounced. Additionally, an opposite tendency was found between the correlation of speech perception and comprehension with surface area, potentially implying the segregation of the different speech processing in the PT area. Intriguingly, within-hemispheric PT functional-structural coupling could be asymmetric, e.g., existing only in one hemisphere but not in the other. For instance, EEG-measured neurophysiological processing of speech perception correlated with neurite density of the left PT but not the right PT, as recently revealed by Ocklenburg and colleagues (*Ocklenburg et al., 2018*). Consistent with this, within-hemispheric PT functional-structural coupling of neurite density was also asymmetric in our study, highlighting the distinct role of PT neurite density in speech processing between the two hemispheres. The distinct roles of left and right PT in speech processing have been well-documented. A number of studies substantiated that PT of the left hemisphere responded more strongly to lexical-semantic and syntactic aspects of sentence processing, whereas the right hemisphere demonstrated a greater involvement in the speech melody (*Albouy et al., 2020*; *Meyer et al., 2002*). These findings are consistent with those reported for the arcuate fasciculus (AF). The left AF has been identified as a crucial structure for language function (*Giampiccolo and Duffau, 2022*; *Zhang et al., 2021*). Disruption to this pathway has been linked to multimodal phonological and semantic deficits (*Agosta et al., 2010*), while injuries in the right AF did not affect language function (*Zeineh et al., 2015*).

The functional-structural coupling of PT asymmetries may be simply driven by the asymmetry of within-hemispheric functional-structural coupling between the left or right PT per se, but this is not the case. The coupling of PT asymmetries could be accompanied by either asymmetric (e.g. speech perception vs. neurite density) or non asymmetric within-hemispheric functional-structural couplings (e.g. speech perception vs. myelin content or neurite orientation dispersion). On the other hand, asymmetric within-hemispheric structure-function association does not necessarily lead to a coupling of PT asymmetries (e.g. speech perception activation vs. surface area). Finally, PT functional and structural asymmetries could be associated, even though there was no within-hemispheric functional-structural coupling for both left and right PTs (e.g. speech comprehension activation vs. surface area). This implied that PT functional and structural asymmetries and their coupling might represent unique structural and functional information, independent of within-hemispheric PT measures. Overall, there was no simple causal relationship between these functional-structural couplings, and they likely

capture distinct aspects of PT functional-structural associations and, therefore, should be investigated separately.

In line with previous observations (*Tzourio-Mazoyer et al., 2019*; *Tzourio-Mazoyer and Mazoyer, 2017*), HG duplication occurred considerably in healthy adults. Regardless of the HG gyrification pattern, both PT functional and structural AI showed an unimodal frequency distribution (*Guadalupe et al., 2022*; *Packheiser et al., 2020*). This contrasts with the well-documented bimodal frequency distribution of functional lateralization for speech production (*Bruckert et al., 2021*; *Knecht et al., 2000*), suggesting that the populational frequency distribution of language lateralization depends on a specific language processing module. Intriguingly, the occurrence of a left HG duplication was accompanied by a lack of the typically observed leftward asymmetry of PT activation, independent of the right HG duplication, which is consistent with previous studies (*Tzourio-Mazoyer et al., 2015*; *Tzourio-Mazoyer et al., 2018*). This highlights a nontrivial role of the left HG duplication in the functional reorganization of auditory language processing between bilateral PTs. In contrast, the HG duplication changes the PT structural asymmetries in a very complex manner, strongly depending on specific structural measures. Therefore, the impact of HG duplication on PT functional lateralization cannot be simply attributed to its impact on PT structural asymmetries. While the influence of the HG duplication on PT asymmetries has been well observed, they are largely overlooked in specific investigations, possible accounting for the result discrepancy between previous studies on PT asymmetries. Notably, no matter considering the HG duplication or not, how the asymmetries of the Heschl's gyrus and planum temporale are related remain largely unknown, and specific investigation on this topic is needed in future studies. On the other hand, the functional-structural coupling of both PT asymmetries and within-hemispheric PT measures were not significantly affected by the HG gyrification pattern. Therefore, HG duplication-induced individual changes in PT measures or their asymmetry do not necessarily lead to changes in the inter-individual association between these measures or asymmetries, suggesting the robustness of these PT functional-structural couplings.

In conclusion, the association between specific PT functional and microstructural asymmetries provides direct empirical support for the contribution of structural asymmetry to functional lateralization of the same cortical area. Moreover, the findings highlight the critical role of microstructural PT asymmetries in auditory-language processing.

## Materials and methods
### Participants
In the present study, all participants of the human connectome project (HCP young adult, S1200 release) were included. The project was reviewed and approved by the Institutional Ethics Committee of Washington University in St. Louis, Missouri. The HCP young adult cohort consists of healthy individuals without neurodevelopmental, neuropsychiatric, or neurologic disorders. All participants signed written informed consent forms. For more details about the data access authority and ethical approval, please refer to *Van Essen et al., 2013*; *Van Essen et al., 2012*.

Due to quality issues (quality control codes A and B from the HCP minimal preprocessing pipeline), 72 subjects were excluded. To control for the potential confounding effect of handedness, we included only qualified right-handed subjects (907 in total, Edinburgh Handedness questionnaire >20) in the analysis.

### Behavioral language test
Two behavioral language tests in the NIH toolbox were applied to HCP individuals: the oral reading recognition test (ORRT, measuring the ability to read decoding) and the picture vocabulary test (PVT, measuring the ability of linguistic comprehension). Given the well-documented role of PT in language processing as well as the previously reported association of PT with language performance (*Blau et al., 2010*; *Liem et al., 2014*; *Tzourio-Mazoyer and Mazoyer, 2017*), whether and how PT measures and their asymmetries correlate with these two language test scores were evaluated. Specifically, the age-adjusted scores for these two language tests were used in the current study.

## MRI acquisition and preprocessing

MRI data of all subjects were collected using the same 3T Siemens Skyra magnetic resonance machine at Washington University in St. Louis with a 32-channel head coil (*Van Essen et al., 2012*). Briefly, T1-weighted images were acquired by using a magnetized rapid gradient-echo imaging (MPRAGE) sequence with the following parameters: repetition time (TR) = 2400 ms, echo time (TE) = 2.14 ms, reversal time (TI) = 1000 ms, flip angle (FA) = 8°, field of view (FOV) = 224 × 224 mm$^2$, voxel size 0.7 mm isotropic. T2-weighted images were acquired using the variable flip angle turbo spin-echo (SPACE) sequence with the following parameters: TR = 3200 ms, TE = 565 ms, field of view (FOV) = 224 × 224 mm$^2$, voxel size 0.7 mm isotropic.

Diffusion-weighted images were acquired with spin-echo EPI with the following parameters: TR = 5520ms, TE = 89.5 ms, field of view (FOV) = 210 × 180 mm$^2$, voxel size 1.25 mm isotropic, 111 slices, 90 directions for each of three shells of b-values (b = 1000, 2000 and 3000 s/mm$^2$) and 18 nondiffusion-weighted (b = 0 s/mm$^2$) volumes.

Task-fMRI scans were acquired under seven tasks: language processing, working memory, incentive processing, relational processing, motor, social cognition, and emotional processing. The language processing task consisted of two runs containing four blocks of an auditory story task and four length-match blocks of an auditory math task. Specifically, during the story blocks, participants were presented with auditory stories adapted from Aesop's fables, followed by a two-alternative forced-choice question about the topic of the story. During the math blocks, participants were presented with auditory arithmetic operations, followed by 'equals' and then two choices. Participants are asked to push a button to select either the first or the second answer. For more detailed paradigms of all these tasks, please refer to *Barch et al., 2013* (*Barch et al., 2013*). All fMRI data were acquired by using a gradient-echo echo planar imaging (GE-EPI) sequence with the following parameters: TR = 720 ms, TE = 33.1 ms, FA = 2°, FOV = 208 × 180 mm$^2$, voxel size 2 mm isotropic, 72 slices, multiband accelerated factor = 8.

All MRI images were preprocessed using the HCP minimal preprocessing pipeline (*Glasser et al., 2013*). For each subject, the HCP minimal preprocessing pipeline provides the native pial and white surfaces that are resampled onto the standard 32 k_fs_LR mesh (~32 k vertices for each hemispheric surface).

## Manually delineating PT and determining the HG gyrification pattern

We followed a well-established procedure for manually delineating PT and determining HG gyrification patterns (*Altarelli et al., 2014*; *Vanderauwera et al., 2018*). The procedure was carried out using the Anatomist software platform embedded in BrainVISA, a sophisticated visualization and labeling tool (*Geffroy et al., 2011*). The Anatomist software platform allows for simultaneously localizing a given coordinate on the surface as well as on the coronal, axial, and sagittal views of the T1 image.

Two well-trained raters (Q.P. and Z.G.) blinded to the subjects' demographics carefully examined the native T1 image for each subject and determined the HG gyrification pattern according to the widely used criterion (*Abdul-Kareem and Sluming, 2008*). As illustrated in *Figure 1*, there were three types of HG gyrification patterns. The first is the sHG, i.e., only one transverse gyrus on the superior temporal gyrus (STG). The second is CSD: the sulcus intermedius of Beck (SI) splits the HG by at least half of the length but never extends to the internal border of the gyrus. The last is complete posterior duplication (CPD): the HG is medially split into two separate parts by an additional Heschl's sulcus (HS1). In the present study, both CSD and CPD were classified as dHG.

Next, the two raters performed the PT delineation on the native pial surface for each subject while simultaneously viewing the coronal, axial, and sagittal slices of the T1 image. According to *Altarelli et al., 2014* (*Altarelli et al., 2014*), the anterior border of the PT was indicated by Heschl's sulcus or the second Heschl's sulcus; the posterior border of the PT was indicated by a change in the slope of the continuous plane characterizing the planum on the coronal view; and the lateral border was defined as the most lateral margin of the STG. For more details, please refer to *Altarelli et al., 2014* and *Vanderauwera et al., 2018* (*Altarelli et al., 2014*; *Vanderauwera et al., 2018*).

## PT functional activation and asymmetry index

The HCP minimal preprocessing pipeline provides individual-level *T* activation maps on the 32 k_fs_LR cortical surface for three contrasts between task conditions ('story – baseline,' 'math – baseline,'

and 'story – math'). For each contrast, we applied the widely-used LI-toolbox approach to quantify PT functional activation and its AI. This approach avoids the dependency on a single threshold for identifying activation and proves robust and specific for computing the AI of functional activation (*Wilke and Lidzba, 2007*; *Wilke and Schmithorst, 2006*). Briefly, a number of thresholds from 0 to the maximum $T$ value of bilateral PTs were applied to the activation map of the left and right region of interest (i.e. PT). At each threshold, the activation$_{left}$, activation$_{right}$, and AI values (i.e. 'activation$_{left}$ – activation$_{right}$ / activation$_{left}$ + activation$_{right}$') are first estimated iteratively using a bootstrap algorithm, in which the activation$_{left}$ or activation$_{right}$ is defined as '(the total vertex area weighted $T$ values across vertices survived the threshold) / (the total number of vertices across the entire region of interest),' a measure taking into account both the relative regional size of activated vertices and their $T$ values. A histogram analysis is then followed to determine the final values for the activation$_{left}$, activation$_{right}$, and AI. Lastly, all estimated final activation$_{left}$, activation$_{right}$ and AI values across different thresholds were weighted by their threshold, yielding the overall activation of each PT and AI values for each subject.

For either activation of each PT or AI values, there are significant correlations across individuals among the three contrasts: 'story – baseline' vs. 'math – baseline' (Left: $R = 0.93$, $P_{FWE}$ <0.001; Right: $R = 0.94$, $P_{FWE}$ <0.001; AI: $R = 0.93$, $P_{FWE}$ <0.001), 'story – baseline' vs. 'story – math' (Left: $R = 0.52$, $P_{FWE}$ <0.001; Right: $R = 0.47$, $P_{FWE}$ <0.001; AI: $R = 0.47$, $P_{FWE}$ <0.001), 'math – baseline' vs. 'story – math' (Left: $R = 0.25$, $P_{FWE}$ <0.001; Right: $R = 0.22$, $P_{FWE}$ <0.001; AI: $R = 0.24$, $P_{FWE}$ <0.001). We then performed exploratory factor analysis on the PT activation$_{left}$, activation$_{right}$, and AI across the three contrasts, separately. Two main factors were consistently obtained, with one loading predominantly on the contrast of 'story – math' (loadings of the activation$_{left}$: 'story – baseline,' 0.39; 'math – baseline,' 0.09; 'story – math,' 0.99; loadings of the activation$_{right}$: 'story – baseline,' 0.31; 'math – baseline,' 0.04; 'story – math,' 0.98; loadings of the AI: 'story – baselin,', 0.34; 'math – baseline,' 0.09; 'story – math,' 0.99), and the other loading predominantly on the contrasts of 'story – baseline' and 'math – baseline' (loadings of the activation$_{left}$: 'story – baseline,' 0.94; 'math – baseline,' 0.99; 'story –math,' 0.21; loadings of the activation$_{right}$: 'story – baseline,' 0.96; 'math – baseline,' 0.99; 'story – math,' 0.21; loadings of the AI: 'story – baseline,' 0.93; 'math – baseline,' 0.99; 'story – math,' 0.14). Given that the contrast of 'story –math' was originally designed to represent semantic processing (*Binder et al., 2011*), the first main factor was, therefore, taken as speech comprehension. The other main factor captures a shared language processing component between the 'story – baseline' and 'math –baseline' and should be dissociated from speech comprehension. Therefore, the second main factor was taken as speech perception, a widely recognized function of PT, and a truly shared language processing component between the 'story – baseline' and 'math –baseline' contrasts. The two-factor scores were then used in subsequent statistical analyses.

In the first specificity analysis, we also applied the same LI-toolbox approach to compute the PT activation$_{left}$, activation$_{right}$, and AI values for the main contrast under the other six fMRI tasks.

In the second specificity analysis, we also applied the same LI-toolbox approach to compute the activation$_{left}$, activation$_{right}$, and AI values of the entire hemisphere for each of the three contrasts under the language task. As above, two-factor scores representing functional activation of speech perception and comprehension of the entire hemisphere were estimated and applied in statistical analyses.

## PT structural measures and asymmetry index

For each subject, we directly obtained cortical maps of surface area, thickness, and T1w/T2w ratio-based myelin content from the HCP minimal preprocessing pipeline. For each delineated PT, the total surface area and averaged thickness and myelin content across PT vertices were calculated. For each subject, we also calculated the NDI and ODI using a diffusion MRI-based neurite orientation dispersion and density imaging (NODDI) approach (*Zhang et al., 2012*). NODDI is a highly effective method for detecting key features of neurite morphology, which employs a tissue model that detects the intracellular, extracellular, and cerebrospinal fluid compartments (*Zhang et al., 2012*). In the gray matter of the cerebral cortex, the NDI is an estimated volume fraction of the intracellular microstructural environment, with higher NDI indicating greater neurite density. The ODI is a measure of the alignment or dispersion of neurite, with higher ODI indicating more dispersed neurite and lower ODIs indicating more aligned neurite (*Jespersen et al., 2012*; *Zhang et al., 2012*). Specifically, voxelwise values for these two measures were first estimated using AMICO (*Daducci et al., 2015*) and then projected onto

the PT vertices using the ribbon mapping method (*Marcus et al., 2011*). For each delineated PT, we averaged the values of the two measures across PT vertices.

For all structural measures, the AI was computed as (Left-Right) / (Left +Right).

## Interrater reliability and test-retest reproducibility

To assess interrater reliability, the two raters both determined the HG gyrification pattern and delineated the PT for the same 20 randomly selected subjects. The results of the HG gyrification pattern reached 100% consistency. Regarding the PT measures, we calculated the intraclass correlation coefficient (ICC). As shown in *Table 1*, the interrater ICC values ranged from 0.85 to 0.99, indicating excellent interrater reliability for these measures.

To evaluate the test-retest reproducibility of both manual operation and brain imaging, one rater (P.Q.) further determined the HG gyrification pattern and delineated the PT for the 43 test-retest HCP subjects who were rescanned (test-retest interval: 1–11 mo). The test-retest results of the HG gyrification pattern also reached 100% consistency. As shown in *Table 1*, the imaging test-retest ICC values ranged from 0.55 to 0.86, indicating excellent test-retest reproducibility.

## Statistical analysis

In terms of the HG gyrification pattern in both hemispheres, all subjects were divided into four groups: bilateral sHGs (L1/R1), left sHG but right dHG (L1/R2), left dHG but right sHG (L2/R1), and bilateral dHGs (L2/R2). The differences in age and sex distribution across the four groups were evaluated using one-way ANOVA and the Kruskal-Wallis test, respectively. For each group, we tested the hemispheric asymmetry of each PT functional activation and structural measure. A linear mixed model (LME) was used, in which 'hemisphere' was a fixed effect and 'individual identity' was a random effect. In the model, age, sex, and total brain volume (*Williams et al., 2022*) were included as covariates. To further evaluate the influence of the HG gyrification pattern on functional and structural measures of the left and right PT, we divided all subjects into two groups for each hemisphere: sHG or dHG. The PT measures of the two groups were then compared using two-sample t-tests after controlling for age, sex, and hemispheric brain volume.

We then evaluated the correlations of PT functional activation, structural measures (2 × 14 in total), and their AIs (2 × 7 pairs in total) with the scores of two behavioral language tests (i.e. ORRT and PVT), wherein age, sex, HG gyrification type, and hemispheric or total brain volume were covariates.

Next, to determine whether PT structural asymmetries relate to functional lateralization at the individual level, we applied a general linear model (GLM) to each pair of functional and structural AIs (2*5 pairs in total). Specifically, the model includes 'functional AI' as the response variable and 'structural AI,' 'group' (i.e. L1/R1, L1/R2, L2/R1, or L2/R2), and 'structural AI × group' as predictor variables, wherein age, sex, and total brain volume were covariates. As described previously (*Chen et al., 2015*; *Engqvist, 2005*; *Zhao et al., 2019*), we first evaluated whether there was a significant 'structural AI × group' interaction, i.e., whether the correlation between functional and structural AIs differed between the 4 groups. If not, the term 'structural AI' was assessed after excluding the interaction term; if yes, post hoc GLM analysis was conducted to assess the term 'structural AI' for each of the four groups.

To test whether the functional-structural AI correlation is related to or even simply driven by the asymmetry of within-hemispheric PT functional-structural correlation between the left and right hemispheres, we further evaluated the correlation of PT functional activation with its structural measures within each hemisphere (4 × 5 pairs in total) and its asymmetry between the two hemispheres. Specifically, for each hemisphere, we applied a GLM with 'functional activation' as the response variable and 'structural measure,' 'group' (i.e. sHG or dHG), and 'structural measure × group' as predictor variables, wherein age, sex, hemispheric brain volume were covariates. Again, we first evaluated whether there was a significant 'structural measure × group' interaction. If not, the term 'structural measure' was assessed after excluding the interaction term; if yes, a post hoc GLM analysis was conducted to assess the term 'structural measure' for each group. For each pair of functional activation and structure measures showing a significant correlation for both the left and right PT, we then applied permutation tests to determine whether the functional-structural correlations differed between bilateral PTs.

For each type of statistical analysis above, the significance level was set as p<0.05 after Bonferroni correction.

## Acknowledgements

We thank Dr. Irene Altarelli for mentoring our PT manual delineation. This work was supported by the National Natural Science Foundation of China (NSFC) [grant numbers T2325006]; National Natural Science Foundation of China (NSFC) [grant numbers 82172016]; the National Natural Science Foundation of China (NSFC) [grant numbers 82021004]; and the Fundamental Research Funds for the Central Universities (Fundamental Research Fund for the Central Universities) [grant numbers 2233200020]. Datasets were provided by the Human Connectome Project, WU-Minn Consortium (Principal Investigators: David Van Essen and Kamil Ugurbil; 1U54MH091657) funded by the 16 NIH Institutes and Centers that support the NIH Blueprint for Neuroscience Research; and by the McDonnell Center for Systems Neuroscience at Washington University.

## Additional information

### Funding

| Funder | Grant reference number | Author |
|---|---|---|
| National Natural Science Foundation of China | T2325006 | Gaolang Gong |
| National Natural Science Foundation of China | 82172016 | Gaolang Gong |
| National Natural Science Foundation of China | 82021004 | Gaolang Gong |
| Fundamental Research Funds for the Central Universities | 2233200020 | Gaolang Gong |

The funders had no role in study design, data collection and interpretation, or the decision to submit the work for publication.

### Author contributions

Peipei Qin, Qiuhui Bi, Formal analysis, Validation, Investigation, Visualization, Methodology, Writing – original draft, Writing – review and editing; Zeya Guo, Formal analysis, Investigation; Liyuan Yang, Xinyu Liang, Junhao Luo, Xiangyu Kong, Yirong Xiong, Formal analysis, Validation; Haokun Li, Peng Li, Formal analysis, Validation, Investigation; Bo Sun, Supervision, Writing – review and editing; Sebastian Ocklenburg, Validation, Investigation, Writing – review and editing; Gaolang Gong, Conceptualization, Supervision, Funding acquisition, Writing – original draft, Project administration, Writing – review and editing

### Author ORCIDs

Junhao Luo  https://orcid.org/0000-0002-3207-1597
Gaolang Gong  https://orcid.org/0000-0001-5788-022X

### Ethics

The dataset used in this study was publicly available and anonymized. Participants provided their informed consent and the study was previously approved by the Washington University Institutional Review Board as part of the Human Connectome Project.

Reviewer #1 (Public review): https://doi.org/10.7554/eLife.95547.3.sa1
Author response https://doi.org/10.7554/eLife.95547.3.sa2

## Additional files

### Supplementary files

• Supplementary file 1. Supplementary tables. (**a**) Group-level hemispheric asymmetry of planum temporale (PT) functional and structural measures. (**b**) The interaction effect of 'PT structural AI ×

Heschl's gyrus (HG) gyrification pattern' on PT nonspeech-related functional asymmetry indexes (AIs) (the first specificity analysis). (**c**) The correlations of PT structural AIs with PT nonspeech-related functional activation AIs after controlling for the HG gyrification pattern (the first specificity analysis). (**d**) The interaction effect of 'PT structural measure × HG gyrification pattern' on PT nonspeech-related functional activation for each hemisphere (the first specificity analysis). (**e**) The correlations of PT structural measures with PT nonspeech-related functional activation for each hemisphere after controlling for the HG gyrification pattern (the first specificity analysis). (**f**) The interaction effect of 'PT structural AI ×HG gyrification pattern' on speech-related functional AIs of the entire hemisphere (the second specificity analysis). (**g**) The correlations of PT structural AIs with speech-related functional AI of the entire hemisphere after controlling for the HG gyrification pattern (the second specificity analysis). (**h**) The interaction effect of 'PT structural measure × HG gyrification pattern' on speech-related functional activation of the entire ipsilateral hemisphere (the second specificity analysis). (**i**) The correlations of PT structural measures with speech-related functional activation of the entire ipsilateral hemisphere after controlling for the HG gyrification pattern (the second specificity analysis). (**j**) The difference in PT functional and structural metrics between groups with single and duplicated HG within each hemisphere (the effect size, Cohen's D).

• Supplementary file 2. Heschl's gyrus (HG) gyrification pattern. (**a**) The Heschl's gyrus (HG) gyrification patterns of subjects included in the present study.

• MDAR checklist

### Data availability

The list of HCP-subject-ID with a morphological group (L1/R1, L1/R2, etc.) is available in the **Supplementary file 2**. The datasets analysed during the current study are available at https://www.human-connectome.org/study/hcp-young-adult. Other data generated during this study are included in the manuscript and supplementary files.

The following previously published dataset was used:

| Author(s) | Year | Dataset title | Dataset URL | Database and Identifier |
|---|---|---|---|---|
| Ugurbil K, Essen D | 2017 | HCP Young Adult | https://www. humanconnectome. org/study/hcp-young- adult | Human Connectome Project, hcp-young-adult |

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
