## [Editor Report · eLife Assessment]

The authors studied the relationship between structural and functional lateralization in the planum temporale region of the brain, whilst also considering the morphological presentation of a single or duplicated Heschl's gyrus. The analyses are **compelling** due to a large sample size, inter-rater reliability, and corrections for multiple comparisons. The associations in this **important** work might serve as a reference for future targeted-studies on brain lateralization.

---

## [Referee Report · Reviewer #1 (Public review)]

Summary:

Qin and colleagues analysed data from the Human Connectome Project on four right-handed subgroups with different gyrification patterns in Heschl's gyrus. Based on these groups, the authors highlight the structure-function relationship of planum temporale asymmetry in lateralised language processing at the group level and next at the individual level. In particular, the authors propose that especially microstructural asymmetries are related to functional auditory language asymmetries in the planum temporale.

Strengths:

The study is interesting because of an ongoing and long-standing debate about the relationship between structural and functional brain asymmetries, and in particular whether structural brain asymmetries can be seen as markers of functional language brain lateralisation.

In this debate, the relationship between Heschl's gyrus asymmetry and planum temporale asymmetry is rare and therefore valuable here. A large sample size and inter-rater reliability support the findings.

Weaknesses:

The authors highlight the microstructural results, but could also emphasise on their interesting macrostructural results.

---

## [Author Response]

The following is the authors’ response to the original reviews.

**Reviewer #1 (Public Review):**
Summary:Qin and colleagues analysed data from the Human Connectome Project on four right-handed subgroups with different gyrification patterns in Heschl's gyrus. Based on these groups, the authors highlight the structure-function relationship of planum temporale asymmetry in lateralised language processing at the group level and next at the individual level. In particular, the authors propose that especially microstructural asymmetries are related to functional auditory language asymmetries in the planum temporale.Strengths:The study is interesting because of an ongoing and long-standing debate about the relationship between structural and functional brain asymmetries, and in particular whether structural brain asymmetries can be seen as markers of functional language brain lateralisation.In this debate, the relationship between Heschl's gyrus asymmetry and planum temporale asymmetry is rare and therefore valuable here. A large sample size and inter-rater reliability support the findings.Weaknesses:In this case of multiple brain measures, it would be important to provide the reader with some sort of effect size (e.g. Cohen's d) to help interpret the results.

Thank you for pointing this out. In the revised version, the effect size, i.e., Cohen's d, has been incorporated into the results (page 8, line 159-160; page 9, line 181-186, supplementary page 14, Table S14).

In addition, the authors highlight the microstructural results in spite of the macrostructural results. However, the macrostructural surface results are also strong. I would suggest either reducing the emphasis on micro vs macrostructural results or adding information to justify the microstructural importance.

In the original manuscript, we highlighted the results of microstructural measures because the correlations between PT microstructural and functional measures were more pronounced both within the hemispheres and in terms of asymmetry, compared with the significant results of surface area. Following your comments here, we now lowered the tone of microstructure results (page 2, line 40; page 14, line 267), and added relevant discussion regarding the macrostructural results in the revised version (page 18, line 363-370; as copied below):

“As for macrostructural measures, the asymmetric PT surface area was also associated with speech comprehension AI. Given that the within-hemispheric coupling tendency between surface and speech comprehension existed only in the left PT, it was possible that the larger surface area of the left PT led to a less recruitment of its right homologous, and therefore the lateralization of functional activity would be more pronounced. Additionally, an opposite tendency was found between the correlation of speech perception and comprehension with surface area, potentially implying the segregation of the different speech processing in the PT area.”

**Recommendations for the authors:**
I have only some comments that I wish to be addressed by the authors:(1) Please always specify "structural" or "functional" asymmetry or lateralisation, as the reader may be confused.

This has been done in relevant places.

(2) Please state that the scale is not the same between the results in Figure 3.

This have been specified, as suggested (see below).

“Notably, we did not standardize these structural measures, so the scales differed between indicators.”

(3) It may be of interest to the reader to learn more about interpretations of how Heschl's gyrus and planum temporale asymmetries are related.

Thank you for this comment. Given that the asymmetry of Heschl's gyrus was not analyzed in the present study, we do not have direct data/results for such an interpretation. Also, we reviewed the literature but found no relevant results on how Heschl's gyrus and planum temporale asymmetries are related. To address this, specific investigation targeting on this topic is needed. This has now been added in the discussion (page 20, line 415-417).

(4) As this manuscript builds somewhat on the Science Advances article by Ocklenburg et al. (2018), it would be important to discuss how this more liberal planum temporale definition might (or might not) affect the results compared to the more conservative planum temporale definition described here.

Yes, the definition of planum temporale varies across studies. Our current manual one is relatively more conservative than the Ocklenburg et al. (2018), in which the planum temporale was automatically derived from the Destrieux atlas. We believe that the definition of the planum temporale likely have non-trivial impact on the results, and our current manual definition with the consideration of the HG duplication should be more reliable and accurate, therefore favored, relative to the other ones. This has been briefly discussed in the revision (page 15-16, line 300-304).

(5) I would like the authors to briefly but critically discuss what exactly the MRI NODDI model measures and how this is interpreted as measuring microstructural properties of tissue.

We now provided relevant information regarding the NODDI measures (page 26, line 552-558; as copied below).

“NODDI is a highly effective method for detecting key features of neurite morphology, which employs a tissue model that detects three microstructural environments: the intracellular, extracellular and cerebrospinal fluid compartments (Zhang et al., 2012). In the grey matter of the cerebral cortex, the neurite density index (NDI) is an estimated volume fraction of the intracellular microstructural environment, with higher NDIs indicating greater neurite density (Jespersen et al., 2010; Zhang et al., 2012). The orientation dispersion index (ODI) is a measure of the alignment or dispersion of neurite, with higher ODIs indicating more dispersed neurite and lower ODIs indicating more aligned neurite (Jespersen et al., 2012; Zhang et al., 2012).”

(6) While not mandatory, I would be interested to read the authors' thoughts on the evolution of such a functional/(micro)structural lateralisation link of the planum temporale, in light of the literature on planum temporale asymmetries in (newborn) non-human primate species.

Thank you for this inspiring suggestion. We have incorporated relevant discussion into the revised version (page 15, line 281-288; as copied below).

“Moreover, there exist evolutionary evidence supporting the role of the PT as an anatomical substrate for language lateralization. For example, the leftward structural asymmetry of the PT have been observed in multiple non-human primates, including chimpanzees, macaques, and baboons (Becker et al., 2024; Gannon et al., 1998; Xia et al., 2019). Particularly, recent studies on baboons further demonstrated that PT structural leftward asymmetry in newborn baboons could predict future development of communicative gestures, implying a key role of PT structural asymmetry in the lateralized communication system for human and non-human brain evolution (Becker et al., 2024, 2021).”

Reference

Becker Y, Phelipon R, Marie D, Bouziane S, Marchetti R, Sein J, Velly L, Renaud L, Cermolacce A, Anton J-L, Nazarian B, Coulon O, Meguerditchian A. 2024. Planum temporale asymmetry in newborn monkeys predicts the future development of gestural communication’s handedness. *Nat Commun* 15:4791. doi:10.1038/s41467-024-47277-6

Becker Y, Sein J, Velly L, Giacomino L, Renaud L, Lacoste R, Anton J-L, Nazarian B, Berne C, Meguerditchian A. 2021. Early Left-Planum Temporale Asymmetry in newborn monkeys (*Papio anubis*): A longitudinal structural MRI study at two stages of development. *NeuroImage* 227:117575. doi:10.1016/j.neuroimage.2020.117575

Gannon PJ, Holloway RL, Broadfield DC, Braun AR. 1998. Asymmetry of Chimpanzee Planum Temporale: Humanlike Pattern of Wernicke’s Brain Language Area Homolog. *Science* 279:220–222. doi:10.1126/science.279.5348.220

Jespersen SN, Bjarkam CR, Nyengaard JR, Chakravarty MM, Hansen B, Vosegaard T, Østergaard L, Yablonskiy D, Nielsen NChr, Vestergaard-Poulsen P. 2010. Neurite density from magnetic resonance diffusion measurements at ultrahigh field: Comparison with light microscopy and electron microscopy. *NeuroImage* 49:205–216. doi:10.1016/j.neuroimage.2009.08.053

Jespersen SN, Leigland LA, Cornea A, Kroenke CD. 2012. Determination of Axonal and Dendritic Orientation Distributions Within the Developing Cerebral Cortex by Diffusion Tensor Imaging. *IEEE Trans Med Imaging* 31:16–32. doi:10.1109/TMI.2011.2162099

Xia J, Wang F, Wu Z, Wang L, Zhang C, Shen D, Li G. 2019. Mapping hemispheric asymmetries of the macaque cerebral cortex during early brain development. *Hum Brain Mapp*. doi:10.1002/hbm.24789

Zhang H, Schneider T, Wheeler-Kingshott CA, Alexander DC. 2012. NODDI: Practical in vivo neurite orientation dispersion and density imaging of the human brain. *NeuroImage* 61:1000–1016. doi:10.1016/j.neuroimage.2012.03.072

**Reviewer #2 (Public Review):**
Summary:The authors assessed the link between structural and functional lateralization in area PT, one of the brain areas that is known to exhibit strong structural lateralization, and which is known to be implicated in speech processing. Importantly, they included the sulcal configuration of Heschl's gyrus (HG), presenting either as a single or duplicated HG, in their analysis. They found several significant associations between microstructural indices and task-based functional lateralization, some of which depended on the sulcal configuration.Strengths:A clear strength is the large sample size (n=907), an openly available database, and the fact that HG morphology was manually classified in each individual. This allows for robust statistical testing of the effects across morphological categories, which is not often seen in the literature.Weaknesses:- Unfortunately, no left-handers were included in the study. It would have been a valuable addition to the literature, to study the effect of handedness on the observed associations, as many previous studies on this topic were not adequately powered. The fact that only right-handers were studied should be pointed out clearly in the introduction or even the abstract.

Thank for pointing this out. We have explicitly specified this in the Abstract and Introduction.

- The tasks to quantify functional lateralization were not specifically designed to pick up lateralization. In the interest of the sample size, it is understandable that the authors used the available HCP-task-battery results, however, it would have been feasible to access another dataset for validation. A targeted subset of results, concerning for example the relationship between sulcal morphology and task-based functional lateralization, could be re-assessed using other open-access fMRI datasets.

Yes, the fMRI task was not specifically designed to evaluate PT functional lateralization, which has been acknowledged in the discussion (page 17, line 330-342). Given the observed small effect size of our current structural-functional relationship, reproducing similar results with other datasets would require a cohort with a large sample size. This would induce a quite labor-intensive work given our current manual protocol for outlining PT and HG for everyone. The lack of validation with independent dataset has been discussed as a limitation in the revised version. We will try to conduct such a validation in future work, likely after developing an automatic pipeline for accurately extracting the PT and HG in the individual space (like the manual outlining protocol).

- The study is mainly descriptive and the general discussion of the findings in the larger context of brain lateralization comes a bit short. For example, are the observed effects in line with what we know from other 'language-relevant' areas? What could be the putative mechanisms that give rise to functional lateralization based on the microstructural markers observed? And which mechanisms might be underlying the formation of a duplicated HG?

Thank you for these insightful comments. As suggested, we strengthened the discussion as below:

“Another possible explanation could be that higher myelin content and larger surface area in left PT potentially indicated more white matter connection with other language-related regions such as Broca’s area, and therefore is more involved in language tasks than its right homolog (Allendorfer et al., 2016; Catani et al., 2005; Giampiccolo and Duffau, 2022).

The distinct roles of left and right PT in speech processing have been well-documented. A number of studies substantiated that PT of the left hemisphere responded more strongly to lexical-semantic and syntactic aspects of sentence processing, whereas the right hemisphere demonstrated a greater involvement in the speech melody (Albouy et al., 2020; Meyer et al., 2002).

These findings are consistent with those reported for the arcuate fasciculus (AF). The left AF has been identified as a crucial structure for language function (Giampiccolo and Duffau, 2022; Zhang et al., 2021). Disruption to this pathway has been linked to multimodal phonological and semantic deficits (Agosta et al., 2010), while injuries in the right AF did not affect language function (Zeineh et al., 2015).”

Regarding the mechanism underlying the formation of a duplicated HG, we did not come up with good thoughts after careful literature review. Also, we feel that this is kind of out of the scope of the present study and therefore did not add more discussion on this topic.

**Recommendations for the authors:**
(1) The data availability statement makes no explicit mention of the manual labels of HG configuration. Would the authors consider making available a list of HCP-subject-ID with a morphological group (L1/R1, L1/R2, etc.) for replicability and for re-use by other researchers?

The list of HCP-subject-ID with a morphological group (L1/R1, L1/R2, etc.) is now available in the supplementary material 2. We have specified this in the revised version.

(2) It would be helpful to state again the statistical tests associated with the p-value in the figure/table caption, e.g. Table 2.

As suggested, we now specified the statistical method in the figure/table caption.

(3) Sometimes, the y-axis labels are missing or not clear, for example in Figure S2.

Sorry about these. We double-checked all the figures, and corrected the missing or unclear labels for Figure S2 and S3 in the revised version.

(4) In a few instances the font sizes vary within a figure caption.

This has been corrected in the revision.

Reference

Agosta F, Henry RG, Migliaccio R, Neuhaus J, Miller BL, Dronkers NF, Brambati SM, Filippi M, Ogar JM, Wilson SM, Gorno-Tempini ML. 2010. Language networks in semantic dementia. *Brain J Neurol*
**133**:286–299. doi:10.1093/brain/awp233

Albouy P, Benjamin L, Morillon B, Zatorre RJ. 2020. Distinct sensitivity to spectrotemporal modulation supports brain asymmetry for speech and melody. *Science*
**367**:1043–1047. doi:10.1126/science.aaz3468

Allendorfer JB, Hernando KA, Hossain S, Nenert R, Holland SK, Szaflarski JP. 2016. Arcuate fasciculus asymmetry has a hand in language function but not handedness. *Hum Brain Mapp*
**37**:3297–3309. doi:10.1002/hbm.23241

Catani M, Jones DK, Ffytche DH. 2005. Perisylvian language networks of the human brain. *Ann Neurol*
**57**:8–16. doi:10.1002/ana.20319

Giampiccolo D, Duffau H. 2022. Controversy over the temporal cortical terminations of the left arcuate fasciculus: a reappraisal. *Brain J Neurol*
**145**:1242–1256. doi:10.1093/brain/awac057

Meyer M, Alter K, Friederici AD, Lohmann G, von Cramon DY. 2002. FMRI reveals brain regions mediating slow prosodic modulations in spoken sentences. *Hum Brain Mapp*
**17**:73–88. doi:10.1002/hbm.10042

Zeineh MM, Kang J, Atlas SW, Raman MM, Reiss AL, Norris JL, Valencia I, Montoya JG. 2015. Right arcuate fasciculus abnormality in chronic fatigue syndrome. *Radiology*
**274**:517–526. doi:10.1148/radiol.14141079

Zhang H, Schneider T, Wheeler-Kingshott CA, Alexander DC. 2012. NODDI: Practical in vivo neurite orientation dispersion and density imaging of the human brain. *NeuroImage*
**61**:1000–1016. doi:10.1016/j.neuroimage.2012.03.072

Zhang J, Zhong S, Zhou L, Yu Yamei, Tan X, Wu M, Sun P, Zhang W, Li J, Cheng R, Wu Y, Yu Yanmei, Ye X, Luo B. 2021. Correlations between Dual-Pathway White Matter Alterations and Language Impairment in Patients with Aphasia: A Systematic Review and Meta-analysis. *Neuropsychol Rev*
**31**:402–418. doi:10.1007/s11065-021-09482-8

**Reviewing Editor:**
I encourage the authors to incorporate the suggestions of the reviewers, such as:(1) to provide more in-depth interpretations about how and why structural and functional lateralization relate,

Done.

(2) to provide statistical effect sizes,

Done.

(3) to make their sulcal-morphology classification openly available,

Done.

(4) to provide statistical effect sizes,

Done

(5) to discuss the possible impact of diverging PT definitions with regard to previous studies,

Done.

(6) to provide more in-depth interpretations about how and why structural and functional lateralization relate.

Done.

Detailed comments:In an impressive cohort of 907 human participants, the present paper presents a very interesting set of data on PT asymmetries not only at the macro-structural but also at the microstructural levels in order to investigate their potential correlates with PT functional asymmetry in relation to perceptual acoustic language tasks.I believe this is a key paper for the following reasons:(1) it provides critical data and results for addressing a controversial but important question: the relevance of measures of anatomical asymmetry for inferring its language-related functional hemispheric specialization;(2) to do so, the authors made a very impressive effort to manually trace the anatomical delineation of the planum temporale at different levels in every participant, the best (but crazy time-consuming) approach so far to document interindividual variability of the PT and to address such a question;(3) the contribution is particularly relevant regarding the statistical power of the study, the study and measures having been done in 907 participants!(4) I also found the study well designed and well written with great relevance of the findings for the field.As the results, the authors reported asymmetric measures of microstructural asymmetry (including intracortical myelin content, neurite density, and neurite orientation) but also of macrostructural asymmetries in relation to functional lateralization for language.Comments：I have only 2 additional minor comments of my own:(1) In agreement with reviewer 2, I don't understand why the authors seem to downplay the links they found between gross PT asymmetry and functional lateralization. I recommend the authors to highlight and discuss this important result, just as the microstructural PT asymmetries and their functional links.

This has been done (page 18, line 363-370).

(2) PT structural asymmetry (both micro & macro) has been well documented in nonhuman primates (and their functional link with manual lateralization for gestural communication). Without detailing this literature, I recommend the authors at least mention this literature as a comparative perspective in the introduction and/or discussion in order to make the question of PT asymmetry less anthropocentric.

This has been done (page 15, line 281-288).